# Thermodynamic Study on Hydrogen Reduction of Germanium Tetrachloride to Germanium

**DOI:** 10.3390/ma17051079

**Published:** 2024-02-27

**Authors:** Dingfang Cui, Zhiying Ding, Tongbo Wang, Bin Kou, Fengyang Chen, Yanqing Hou, Bin Yang, Gang Xie

**Affiliations:** 1Faculty of Metallurgical and Energy Engineering, Kunming University of Science and Technology, Kunming 650093, Chinazyayaqinene@163.com (Z.D.); chenfengyang0315@163.com (F.C.); yang18108818480@163.com (B.Y.); 2Yunnan Chihong International Germanium Co., Ltd., Qujing 655000, China; wtb083@163.com (T.W.); koubin97@163.com (B.K.); 3Chinalco Research Institute of Science and Technology Co., Ltd., Beijing 102209, China; 4State Key Laboratory of Advanced Metallurgy for Non-Ferrous Metals, Kunming 650051, China

**Keywords:** germanium, germanium tetrachloride, thermodynamics, germanium deposition

## Abstract

This study elucidates the thermodynamic reaction mechanism of the GeCl_4_ hydrogen reduction process for Ge preparation. Five independent reactions in the Ge-Cl-H ternary system were identified, utilizing the phase law, mass conservation principles, and thermodynamic data, with H_2_ as the reducing agent. Additionally, the effects of the temperature, feed ratio, and pressure on the germanium deposition rate during the GeCl_4_ hydrogen reduction process were investigated, guided by these five reactions. The results indicate that, with fixed temperature and pressure, a higher feed ratio ($n_{H_{2}}/n_{{GeCl}_{4}}$) leads to an increased germanium deposition rate. Conversely, with a constant feed ratio, increased pressure results in a lower deposition rate at low temperatures. The optimal operating conditions for germanium preparation via the hydrogen reduction of GeCl_4_ were determined: the temperature was 450 °C, the feed ratio was 20, the pressure was 0.1 MPa, and the deposition rate of the germanium was 36.12% under this condition.

## 1. Introduction

Germanium, a crucial strategic metal resource in the twenty-first century, is widely used in detectors [1], optics [2], microelectronics [3], and solar cells [4,5,6] due to its excellent infrared transmittance, electromagnetic refractive index, and photoelectric conversion efficiency [7,8,9]. The primary methods for germanium production include: (1) chemical vapor deposition, involving deposition on substrates using hydrogen-diluted germanium alkane [10] or digermanium alkane [11]; (2) disproportionation, synthesizing germanium from germanium diiodide (GeI_2_) [12]; (3) the “chlorination–hydrolysis-reduction” process [13,14,15], which starts with hydrolyzing germanium tetrachloride (GeCl_4_) to germanium dioxide (GeO_2_), followed by hydrogen (H_2_) reduction and zone melting for purifying high-purity germanium. However, these traditional methods are hindered by low efficiency, the potential introduction of impurities leading to secondary pollution, lengthy processes, and high equipment costs. Consequently, there is a pressing need to explore more efficient purification techniques.

The industry widely employs a chloride method for obtaining high-purity germanium. The main problem of the chloride technology is the low yield of germanium (≤70%), and also the substantial loss with chloride sewage water and the contamination of the final product at the stage of germanium tetrachloride hydrolysis. In addition, chlorides are toxic and corrosion-active substances, which makes the “chloride technology” rather labor-consuming gas as regards its instrumentation. Up to 70% of capital expenditure and maintenance expenses are constituted by costs associated with the purification of wastewater and effluent gases, as well as the appreciation of the equipment [3,8]. However, all chemical methods for the synthesis of hydrides have important shortcomings. First, they are nonselective: hydrides of other elements are simultaneously generated, which requires the application of complex and expensive purification techniques [6,12], and the resulting gas contains substantial amounts of by-released hydrogen. Second, the reaction yields a large amount of toxic waste, with which a part of germanium is lost. Third, the process performed in the reactor is difficult to control and monitor and, therefore, a search for other ways to synthesize germanium, based on research into the mechanisms of reaction product formation, is of particular interest. In the development of new synthesis methods, it is necessary to obtain the minimum amount of starting reagents and the number of synthesis stages, which makes the reduction in the contamination level [9,12] of the product possible. It seems that the most advantageous in this regard is the direct catalytic reduction of silicon and germanium tetrachloride to obtain as final products.

The preparation of germanium via the hydrogen reduction of germanium tetrachloride, involving the direct reduction with vaporized GeCl_4_ in a reactor under high-speed H_2_ flow, minimizes impurity introduction and significantly streamlines the process. Numerous scholars worldwide have conducted extensive research on germanium preparation through the hydrogen reduction of germanium tetrachloride. This research primarily centers on two aspects: the catalytic hydrogen reduction of liquid germanium tetrachloride for germanium nanomaterial preparation, which lowers reaction temperatures and enhances germanium conversion rates through catalyst addition. For instance, Kadomtseva et al. [16] enhanced the hydrogenation of germanium tetrachloride by adding copper nanoparticles to modified multiwalled carbon nanotubes, proposing a detailed reaction mechanism for the catalyzed reduction. Vorotontsev et al. [17] utilized density functional theory to explore the elevated catalytic activity of metals, like Ni, Pt, and W, in dissociative hydrogen chemisorption reactions. Kadomtsevaa et al. [18] conducted hydrogen reduction experiments on germanium tetrachloride using NiCl_2_ as a catalyst, demonstrating a reduction in process temperature from 1123 K to 623 K and a 43 KJ/mol decrease in activation energy compared to the uncatalyzed reaction. Heath [19] prepared germanium nanowires from liquid germanium tetrachloride through hydrogen reduction under high-pressure conditions, yielding diameters from 7 to 30 nm and lengths up to 10 μm. Kadomtseva [20] developed a tungsten-catalyzed hydrogen reduction process for germanium tetrachloride, effectively lowering the reaction temperature and simplifying the germanium preparation steps. The hydrogen reduction of gaseous germanium tetrachloride, used in preparing germanium thin films, enhanced the deposition rates and uniformity through surface depressurization and nonthermal plasma techniques. Ishii et al. [21] demonstrated that, in the CVD deposition of Ge thin films on Ge substrates using a GeCl_4_-H_2_ system, the surface reaction involved dissociated adsorbed hydrogen atoms and surface-adsorbed GeCl_2_ molecules instead of a hydrogen gas-phase reduction. Gresback et al. [22] employed a nonthermal plasma method to synthesize monodispersed germanium nanocrystals for spin-coating, preparing Ge thin films in the GeCl_4_-H_2_ system. Chen et al. [23] used plasma-enhanced chemical vapor deposition with GeCl_4_ as the precursor to deposit high-quality Ge thin films on silicon substrates, ideal for Ge/Si photodiode fabrication.

Seeing this impressive precedent work has inspired the current study toward the evaluation of the reduction of germanium tetrachloride to germanium. However, while existing studies primarily focus on the kinetics of the germanium tetrachloride hydrogen reduction process to unravel its reaction mechanism, the thermodynamic aspects of the process have received minimal attention. The thermodynamic influence on the entire reaction process is significant. Investigating thermodynamics can elucidate the relationship between the deposition conditions (such as the temperature, pressure, and inlet ratio) and the deposition rate in the GeCl_4_-H_2_ system, aiding in identifying the optimal operating conditions and enhancing the primary conversion rate. This paper delves into the thermodynamics of germanium tetrachloride hydrogen reduction for germanium preparation, examining the impacts of temperature, the feed ratio, and pressure on the deposition rate. The analysis aims to determine the optimal operating conditions, offering valuable insights for industrial application.

## 2. Analysis of Ge-H-Cl Ternary System

According to the phase rule, the heterogeneous reaction of the Ge-H-Cl ternary system has three degrees of freedom. During the whole reaction process of the germanium tetrachloride hydrogen reduction for the preparation of germanium, no H- or Cl-containing substances are generated in the liquid or solid phase, so that the Cl/H in the gas phase remains constant throughout the reaction process, which is determined by the ratio of the feed gas, and the Cl/H of the feed gas can be used as one of the variables. Therefore, under certain conditions of temperature, pressure, and Cl/H, when the reaction reaches equilibrium, the content of each component can be obtained, and thus the Ge/Cl ratio at equilibrium can be calculated, denoted as ${\left(Ge/Cl\right)}_{eq}$. The Ge/Cl ratio of the feed gas can be calculated by the feed gas, denoted as the ${\left(Ge/Cl\right)}_{feed}$, and thus the germanium yield (*η*) can be calculated by Equation (1):(1)$$\eta =\frac{{\left(Ge/Cl\right)}_{feed}-{\left(Ge/Cl\right)}_{eq}}{{\left(Ge/Cl\right)}_{feed}}$$

Based on the thermodynamic data of the relevant reactions, the molar fractions of the components, ΔG_m_, K, and the conversion rate from germanium chloride are studied, and the results obtained can be applied to the actual production process to optimize the conditions. Based on the analysis of thermodynamics and actual chemical substances, there can be eight components, which are the main components, in the reaction system for the preparation of germanium by the hydrogen reduction of germanium tetrachloride, such as GeCl_4_ (g), GeCl_3_ (g), GeCl_2_ (g), GeCl (g), GeH_4_ (g), H_2_ (g), HCl (g), and Ge (s). The other components are very scarce and can be ignored.

Therefore, the number of species (M) considered in the preparation of germanium by hydrogen reduction of germanium tetrachloride is 8, and the process is a nonhomogeneous reaction of the Ge-H-Cl ternary system, so the number of independent components (N) is 3, which leads to the number of independent reactions: M-N = 5. Thus, there are 5 independent reactions in the process. A group of independent reactions can be obtained by analyzing as follows:GeCl_4_(g) + 2H_2_(g) = Ge + 4HCl(g)    (primary reaction) (2)
2GeCl_4_(g) + 3H_2_(g) = GeCl(g) + 6HCl (3)
GeCl_4_(g) + H_2_(g) = GeCl_2_(g) + 2HCl(g) (4)
2GeCl_4_(g) + H_2_(g) = GeCl_3_(g) + 2HCl(g) (5)
GeCl_4_(g) + 4H_2_(g) = GeH_4_(g) + 4HCl(g) (6)

Based on Brinklev’s method, only five reactions are independent in the GeC1_4_ hydrogen system, while the other reactions are a linear combination of five independent reactions. In the process of converting GeCl_4_ to Ge, only the primary reaction generates solid germanium, while the rest of the reactions are gas-phase reactions. In addition, the main components in the reaction are GeCl_4_ (g), GeCl_3_ (g), GeCl_2_ (g), GeCl (g), GeH_4_ (g), H_2_ (g), HCl (g), and Ge (s). Therefore, the reactions in Equations (2)–(6) can be selected as a set of independent reactions, and the relationship between ΔG_m_ and temperature can be fitted based on the thermodynamic data of the five independent chemical reactions listed in the GeCl_4_ hydrogen reduction system in Table 1, as shown in Figure 1a.

As shown in Figure 1a, the ΔG_m_ values of most reactions are more significant than zero in the range of 200–450 °C, indicating that the equilibrium constant (K) value is minimal and the degree of the reaction is minimal. After exceeding 450 °C, the ΔG_m_ values of the primary reaction (2) and the side reaction (4) are both less than 0, indicating that the equilibrium constant (K) increases and the degree of reaction progression is much greater than the other three reactions. According to the thermodynamic data in Table 2, the relationship between the equilibrium constant K and temperature for five independent chemical reactions in the GeCl_4_ hydrogen reduction system is fit, as shown in Figure 1b.

The K values of the five independent chemical reactions are small in the 200–600 °C range. When the temperature is higher than 550 °C, the K values of the side reactions (3), (5), and (6) increase slightly, and the K values of the main reaction (2) and the side reaction (4) increase rapidly, and the K value of the side reaction (4) is more significant than that of the primary reaction (2). At 600 °C, the K value of primary reaction (2) was 4.21, and the K value of side reaction (4) was 9.38; at 800 °C, the K value of primary reaction (2) was 8.96, and the K value of side reaction (4) was 28.7. With the increase in temperature, the K value of both reactions increased, but the growth rate of the side reaction (4) K value was more significant than that of the primary reaction (1). Therefore, the operating temperature should be controlled between 600 and 800 when the K value of the primary reaction (2) is more significant, and the difference between the K value of the main reaction (2) and the secondary reaction (4) is not very large.

### 2.1. Equilibrium Component Analysis

The equilibrium gas-phase components can visualize the content of each gas-phase component when the reaction reaches equilibrium, thus providing a reliable basis for the treatment of the tail gas. In this section, the changes in each gas-phase component with temperature when the reaction reaches equilibrium are investigated under feed ratios of 5, 15, and 25 and pressures of 0.1, 0.2, and 0.5 MPa, respectively. The equilibrium mole amount (*x_i_*) for the gas-phase substance *i*, such as GeCl_4_, GeCl_3_, GeCl_2_, GeCl, H_2_, and HCl, were mainly analyzed, and xi was the mole fraction of the *i*th substance when the reaction reached equilibrium. The results are shown in Figure 2, Figure 3 and Figure 4.

It can be seen from Figure 2, Figure 3 and Figure 4 that the trend of changes in the equilibrium gas-phase components is similar. As shown in Figure 2b, with the increase in temperature, the content of H_2_ is slightly decreased from 14.9 mol to 14.0 mol, the content of GeCl_4_ has a significant decrease from 0.94 mol to 2.57 × 10^−5^ mol, the content of GeCl_3_ first increases from 2.45 × 10^−3^ mol to 7.79 × 10^−2^ mol, then it decreases to 4.67 × 10^−3^ mol, and the HCl, GeCl_2_, and GeCl content increases significantly. In addition to H_2_, the main component at low temperatures is GeCl_4_, and when the temperature increases, the main component changes to HCl and GeCl_2_. When the pressure is 0.1 MPa, 0.2 MPa, or 0.5 MPa, the equilibrium gas-phase component change trend is basically similar, but there is a difference: with the increase in the pressure, the intersection of the curves moves to the right. As shown in Figure 3, when the pressure is 0.1 MPa, the intersection point of the curves GeCl and GeCl_3_ is between 800 and 900 °C. However, when the pressure is 0.5 MPa and the temperature is 900 °C, the content of GeCl is 3.46 × 10^−3^ mol, and GeCl_3_ is 6.18 × 10^−3^ mol. Under this condition, there is no intersection between the two curves. According to the trend of the changes in the equilibrium gas-phase components, these two curves will intersect as the temperature continues to increase. This indicates that, when the reaction reaches equilibrium, the content of GeCl_4_ changes more slowly with the increase in the pressure, so increasing the pressure is unfavorable to the germanium deposition process.

When the initial feed ratio is increased from 5 to 15 to 25, the content of GeCl_4_ is significantly reduced at the same temperature, which means that increasing the feed ratio can increase the germanium deposition rate. When the temperature is 600 °C, the content of GeCl_4_ is reduced from 0.022 mol to 1.4 × 10^−3^ mol. Therefore, in the actual production process, excessive hydrogen is necessary. The above analysis shows that high temperature, excessive hydrogen, and relatively low pressure are beneficial to improving the germanium yield. The high temperature and excessive hydrogen can improve the germanium deposition rate. Compared to higher pressure, lower pressure decreases the temperature when achieving the same germanium conversion rate. It reduces the energy consumption.

### 2.2. Effect of Temperature on Germanium Deposition Rate

When the feed ratio and pressure were kept constant, the effect of the feed ratio on the germanium deposition rate was analyzed. The effect of temperature on the germanium deposition rate was analyzed under the conditions of feed ratios of 10, 15, and 20, and pressures of 0.1 MPa, 2 MPa, and 0.5 MPa, respectively, and temperatures in the range from 200 °C to 900 °C, as shown in Figure 5 and Figure 6.

From Figure 5, it can be seen that, under the condition of the same feed ratio, when the temperature is 350~500 °C, the germanium deposition rate is greatly affected by the pressure, and the higher the pressure, the lower the germanium deposition rate. When the temperature is lower than 350 °C, the germanium deposition rate is meager, and the change in the pressure almost does not affect the germanium deposition rate. When the feed ratio is 10 and p is 0.1, 0.2, and 0.5 MPa, the germanium deposition rate is 3.96%, 3.34%, and 2.64%, respectively. This shows that, when the temperature is lower than 350 °C, the pressure change has almost no effect on the germanium deposition rate. The main reason for this phenomenon is that, when the temperature is lower, the log K values of both the primary reaction (2) and the side reactions (3)–(6) are negative, i.e., the equilibrium constant K is less than one. From Figure 6, it can be noticed that the deposition rate of germanium increased significantly with the increase in the feed ratio under the same pressure condition. Therefore, it is clear that it is essential to control the temperature in the optimum range during germanium preparation by the hydrogen reduction of germanium tetrachloride. The optimum range of temperature is 350~500 °C. When the feed ratio is 20 and the pressure P is 0.1, 0.2, and 0.5 MPa, the optimal temperature is 450, 460, 480 °C, and the germanium deposition rate is 36.12, 35.12, and 32.27%, respectively.

### 2.3. Effect of Feed Ratio on Germanium Deposition Rate

The effect of the feed ratio on the germanium deposition rate was analyzed when the pressure and temperature were kept constant. The variation in the germanium deposition rate with feed ratios in the range of 5~30 at temperatures of 400 °C, 450 °C, and 500 °C, and pressures of 0.1 MPa, 0.2 MPa, and 0.5 MPa are displayed in Figure 7 and Figure 8, respectively.

From Figure 7 and Figure 8, it can be concluded that, when the pressure and temperature are kept constant, the germanium deposition rate increases gradually as the feed ratio increases, indicating that an excess of hydrogen is necessary in the germanium deposition process. According to Figure 7, the growth rate of the germanium deposition rate is gradually slowed when the feed ratio increases. At a pressure of 0.1 MPa and a temperature of 450 °C, the germanium deposition rate increased by 11.38% when the feed ratio was increased from 10 to 15; the germanium deposition rate increased by 8.99% when the feed ratio was increased from 15 to 20; the germanium deposition rate only increased by 7.19% when the feed ratio was increased from 20 to 25. This indicates that the growth rate of germanium deposition gradually decreases while the feed ratio increases. When the feed ratio increases to a specific value, further increasing the feed ratio has little effect on the germanium deposition rate. At the same time, a high feed ratio will increase the content of H_2_ in the exhaust gas, thereby increasing the difficulty of separating and treating H_2_ in the exhaust gas, resulting in increased production costs.

Above all, it can be learned that excessive hydrogen is necessary for the process of the hydrogen reduction of germanium tetrachloride to prepare germanium, but the feed ratio should be reasonable. Otherwise, it will lead to the following problems: (1) increasing the feed ratio has little effect on the germanium deposition rate; (2) increasing the feed ratio also increases the HCl content in the product. For example, under the conditions of 0.1 MPa and 450 °C, with a feed ratio of 10, the HCl yield is 1.84; when the feed ratio is 15, the HCl output is 2.22; (3) increasing the feed ratio will increase the difficulty of separating hydrogen from HCl and increase the costs. Therefore, in actual industrial production, the germanium tetrachloride hydrogen reduction preparation of germanium should be controlled with a feed ratio of about 20.

As seen in Figure 8, under the condition of constant temperature, there exists a specific value of the feed ratio; when the feed ratio is more significant than this value, the germanium deposition rate decreases with the increase in the pressure; when the feed ratio is smaller than this value, the germanium deposition rate increases with the increase in the pressure.

### 2.4. Effect of Pressure on Germanium Deposition Rate

The effect of pressure on the germanium deposition rate was analyzed when the temperature and feed ratio were fixed. The changes in the germanium deposition rate in the range of 0.1 MPa~1 MPa at temperatures of 350 °C, 450 °C, and 500 °C, and feed ratios of 10, 15, and 20, respectively, are shown in Figure 9 and Figure 10.

It is evident from Figure 9 that, when the temperature is 500 °C and the feed ratio is certain, the germanium deposition rate is almost unchanged with the increase in the pressure, which is approximately kept as a constant value. When the temperature is lower than 500 °C, the decrease in the germanium deposition rate can be seen with increased pressure. It shows that the increase in pressure will lead to a decrease in the germanium deposition rate, and in the production process, the pressure should be controlled to 0.1 MPa. It is shown in Figure 10 that, as the feed ratio increases, the temperature range that is not affected by pressure also increases. When the feed ratio is 20 and the temperature is 500 °C~700 °C, increasing the pressure does not affect the germanium deposition rate.

According to the above analysis, it can be concluded that, under the optimal feed ratio of 20, when the pressure *p* = 0.1 MPa, the optimal temperature is 450 °C, and the germanium deposition rate is 36.12%; when the pressure *p* = 0.2 MPa, the optimal temperature is 460 °C, and the germanium deposition rate is 35.16%; when the pressure *p* = 0.5 MPa, the optimal temperature is 480 °C, and the germanium deposition rate is 32.27%. From Section 2.4, it can be found that the increase in the pressure will lead to a decrease in the germanium deposition rate, so the pressure should be controlled to 0.1 MPa during the production process. In summary, the best practical production scheme for preparing germanium by the hydrogen reduction of germanium tetrachloride was obtained at the temperature T = 450 °C, a feed ratio 20, and a pressure *p* = 0.1 MPa. The highest Ge deposition rate at this moment is 36.12%.

## 3. Conclusions

This study investigated the thermodynamic reaction mechanism in the process of germanium preparation by the hydrogen reduction of germanium tetrachloride. Based on the thermodynamic analysis, five independent reactions were obtained, the effects of the temperature, feed ratio, and pressure on the deposition rate of germanium in the process were analyzed, and the optimal production conditions of the process were determined. The results are as follows:(1)Relevant thermodynamic data were applied to study the complex chemical reactions of the Ge-H-Cl ternary system in the hydrogen reduction process of germanium tetrachloride, and five independent reactions in the hydrogen reduction process of germanium tetrachloride were identified and plotted with ΔGm-T and logK-T diagrams. In the temperature range from 350 °C to 550 °C, the ΔGm values of the primary reaction (2) and the side reaction (4) are less than 0, and the reaction proceeds efficiently. At high temperatures, the deposition rate of germanium is low because the K value of the side reaction proliferates; at low temperatures, the K value of the primary reaction is minimal, and the reaction proceeds to a low degree. Therefore, in actual production, the low deposition rate of germanium is a normal phenomenon.(2)The germanium deposition rate increases significantly with increasing temperature, and when the temperature exceeds the optimum temperature, the germanium deposition rate decreases with increasing temperature. When the temperature is lower, the germanium deposition rate is meager, and the change in pressure hardly affects the germanium deposition rate. When the temperature is higher, the germanium deposition rate increases with the increase in the feed ratio. The analysis shows that the optimum operating temperature is 450 °C when the pressure is determined to be 0.1 MPa.(3)An excess of hydrogen is necessary in the germanium deposition process. As the feed ratio increases, the germanium deposition rate also increases, but the growth rate of the germanium deposition gradually decreases. When the feed ratio increased to a specific value, the effect of further increase on the germanium deposition rate was negligible. Moreover, when the feed ratio was too large, it increased the production of HCl, simultaneously increasing the cost and difficulty of separating hydrogen in the exhaust gas. Consequently, from the above analysis, it can be seen that, in the actual industrial production, germanium tetrachloride should be controlled to the feed ratio of about 20 in the process of germanium preparation by hydrogen reduction.(4)Within the optimal temperature range, the germanium deposition rate decreases with the increase in pressure. When the temperature was higher, the pressure had almost no effect on the germanium deposition rate, and with the increase in the feed ratio, the temperature range where the pressure did not affect the germanium deposition rate became more considerable. Therefore, in the actual production process, the pressure should be controlled at 0.1 MPa.(5)The optimum practical production conditions for preparing germanium by the hydrogen reduction of germanium tetrachloride were: temperature T = 450 °C, feed ratio 20, pressure *p* = 0.1 MPa. The deposition rate of germanium at this time was 36.12%.

## Figures and Tables

**Figure 1 materials-17-01079-f001:**
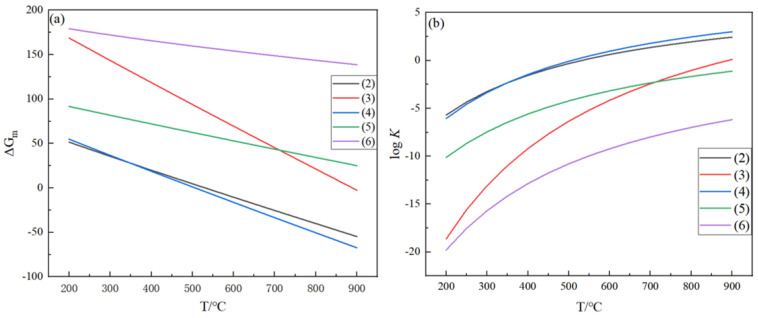
(**a**) Relationship between the Gibbs energy and temperature; (**b**) the relationships of the equilibrium constant and temperature for reactions (2)–(6).

**Figure 2 materials-17-01079-f002:**
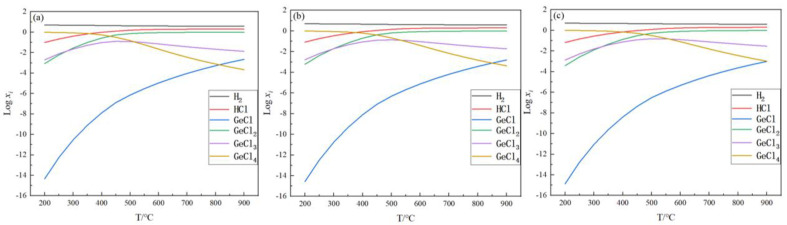
Variation in equilibrium gas-phase components with temperature at $n_{H_{2}}/n_{{GeCl}_{4}}$ = 5, *p* = 0.1 MPa (**a**), 0.2 MPa (**b**), and 0.5 MPa (**c**).

**Figure 3 materials-17-01079-f003:**
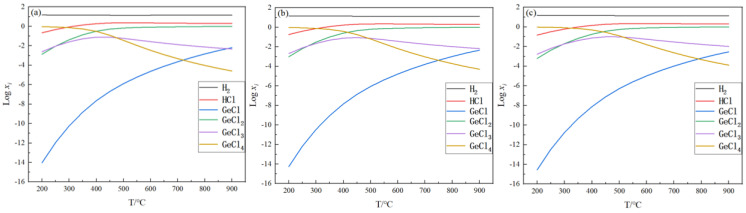
Variation in equilibrium gas-phase components with temperature at $n_{H_{2}}/n_{{GeCl}_{4}}$ = 15, *p* = 0.1 MPa (**a**), 0.2 MPa (**b**), and 0.5 MPa (**c**).

**Figure 4 materials-17-01079-f004:**
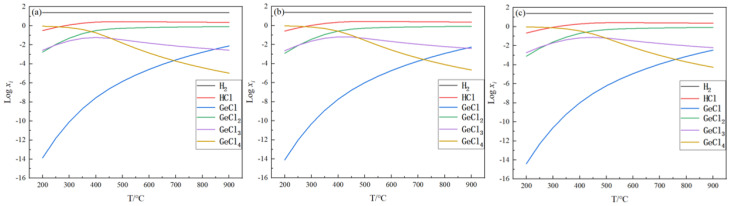
Variation in equilibrium gas-phase components with temperature at $n_{H_{2}}/n_{{GeCl}_{4}}$ = 25, *p* = 0.1 MPa (**a**), 0.2 MPa (**b**), and 0.5 MPa (**c**).

**Figure 5 materials-17-01079-f005:**
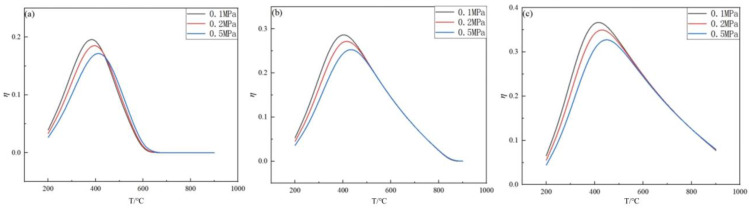
Variation in the Ge deposition rate with temperature for $n_{H_{2}}/n_{{GeCl}_{4}}$ = 10 (**a**), 15 (**b**), and 20 (**c**).

**Figure 6 materials-17-01079-f006:**
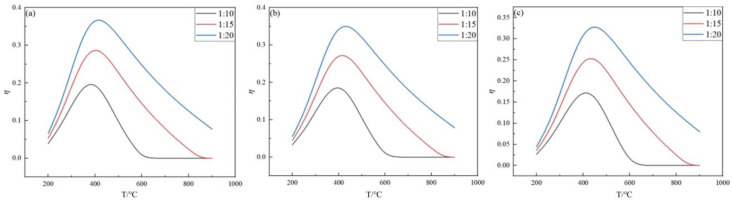
Variation in the Ge deposition rate with temperature at *p* = 0.1 MPa (**a**), 0.2 MPa (**b**), and 0.5 MPa (**c**).

**Figure 7 materials-17-01079-f007:**
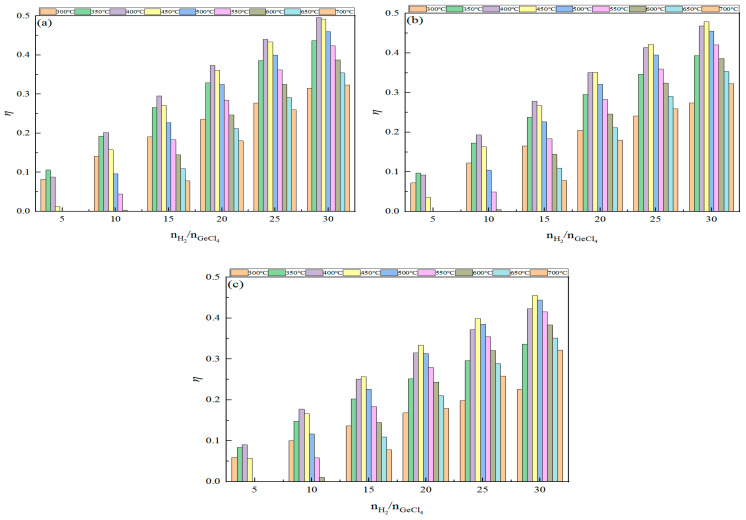
Variation in the Ge deposition rate with feed ratios at *p* = 0.1 MPa (**a**), 0.2 MPa (**b**), and 0.5 MPa (**c**).

**Figure 8 materials-17-01079-f008:**
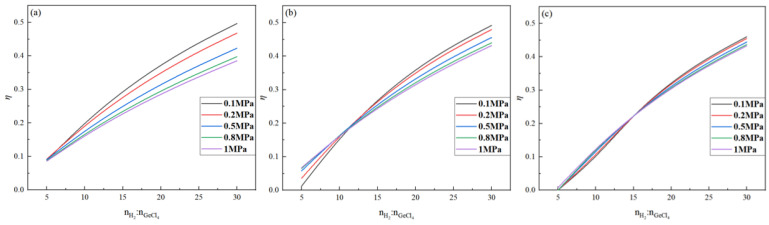
Variation in the Ge deposition rate with the feed ratio and pressure at T = 400 °C (**a**), 450 °C (**b**), and 500 °C (**c**).

**Figure 9 materials-17-01079-f009:**
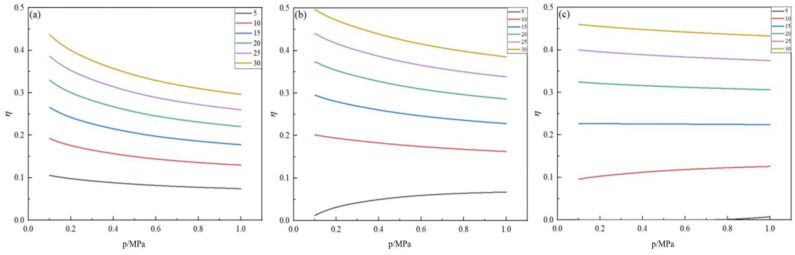
Variation in the Ge deposition rate with pressure at T = 350 °C (**a**), 450 °C (**b**), and 500 °C (**c**).

**Figure 10 materials-17-01079-f010:**
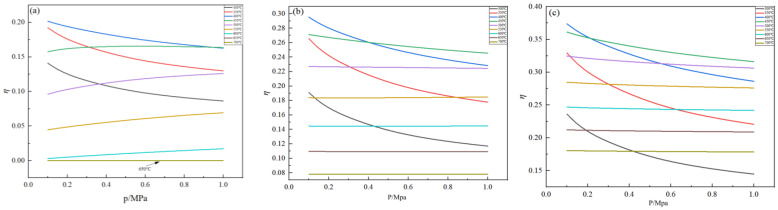
Variation in the Ge deposition rate with pressure for $n_{H_{2}}/n_{{GeCl}_{4}}$ = 10 (**a**), 15 (**b**), and 20 (**c**).

**Table 1 materials-17-01079-t001:** The Gibbs free energy ΔG_m_ change in reactions (2)–(6).

T/°C	ΔG_m_/kJ				
	Reaction (2)	Reaction (3)	Reaction (4)	Reaction (5)	Reaction (6)
200	51.524	168.76	54.852	91.678	179.172
250	43.509	156.087	45.716	86.719	175.581
300	35.585	143.502	36.662	81.801	172.16
350	27.744	130.998	27.683	76.92	168.885
400	19.979	118.57	18.774	72.074	165.739
450	12.285	106.212	9.931	67.261	162.705
500	4.654	93.918	1.148	62.477	159.771
550	−2.919	81.685	−7.578	57.722	156.923
600	−10.44	69.507	−16.253	52.992	154.153
650	−17.915	57.382	−24.88	48.287	151.45
700	−25.348	45.304	−33.462	43.604	148.806
750	−32.745	33.271	−42.002	38.941	146.215
800	−40.109	21.279	−50.502	34.299	143.669
850	−47.444	9.327	−58.967	29.674	141.163
900	−54.754	−2.59	−67.397	25.067	138.693

**Table 2 materials-17-01079-t002:** The equilibrium constant Log K change in reactions (2)–(6).

T/°C	Log K				
	Reaction (2)	Reaction (3)	Reaction (4)	Reaction (5)	Reaction (6)
200	−5.689	−18.632	−6.056	−10.122	−19.782
250	−4.345	−15.586	−4.565	−8.659	−17.533
300	−3.243	−13.079	−3.341	−7.456	−15.691
350	−2.326	−10.982	−2.321	−6.448	−14.158
400	−1.55	−9.202	−1.457	−5.593	−12.862
450	−0.887	−7.673	−0.717	−4.859	−11.754
500	−0.314	−6.346	−0.078	−4.221	−10.795
550	0.185	−5.184	0.481	−3.663	−9.959
600	0.625	−4.159	0.972	−3.17	−9.223
650	1.014	−3.247	1.408	−2.732	−8.570
700	1.361	−2.432	1.796	−2.341	−7.988
750	1.672	−1.699	2.144	−1.988	−7.465
800	1.952	−1.036	2.458	−1.67	−6.994
850	2.207	−0.434	2.743	−1.38	−6.566
900	2.438	0.115	3.001	−1.116	−6.176

## Data Availability

Data are contained within the article.

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
