# Peer review of "Thermodynamic Study on Hydrogen Reduction of Germanium Tetrachloride to Germanium"

_materials, 2024, doi:10.3390/ma17051079_

Round 1

Reviewer 1 Report

Comments and Suggestions for Authors

This study is important, considering germanium is a critical mineral with many areas of use. The introduction clearly explains the need for the study, and the results are also clearly explained. The methodology could be more detailed. Please consider being more detailed in that section. Moreover, please explain what Figure 1 (b) indicates. The figure caption seems cut short. Also, Figures 7 and 8 are difficult to read. Please improve their quality and make the legends and axis bigger. 

-The authors should discuss the differences between Figs 2, 3, and 4 with numbers rather than saying "increase and decrease". The changes in species' behavior under different conditions are small to notice by just looking at the graph. Numbers could explain the phenomenon better. 

-Also, the H2 line is missing in Figs 3 and 4 or too engaged with the other lines that it is not easy to follow. -What would be the conditions if the authors decide on an optimum process? It seems like the changes in some parameters have very minimal impact.  

Comments on the Quality of English Language

Pretty good. No major issues were detected. 

Author Response

Response Letter

Dear Reviewers:

Thank you for your letter and for the comments concerning our manuscript entitled Thermodynamic Study on Hydrogen Reduction of Germanium tetrachloride to Germanium by Dingfang Cui et al., manuscript number: materials-2782978. Those comments are all valuable and very helpful for revising and improving our paper, as well as the important guiding significance to our researches. We have studied comments carefully and have made correction which we hope meet with approval. Revised portion are marked with different colors in the paper.

We sincerely hope that, with these revisions, the manuscript could meet the standard for publication in Materials

Sincerely yours,

Yanqing Hou

Faculty of Metallurgy and Energy Engineering

Kunming University of Science and Technology

Kunming 650093, China

Tel: 86-15987198926

Reviewer 2 Report

Comments and Suggestions for Authors

materials-2782978

1.    In the present manuscript, the authors presented the thermodynamics study on H2-reduction of germanium tetrachloride to zero-valence germanium. Although the chosen topic is good to study, the manuscript lacks in scientific discussion. The illustration of thermodynamics pictures for Gibbs free energy versus temperature is not a big deal, it can be easily done using the software available. However, the explanation and presentation of data is expected from the authors. Unfortunately, the authors failed to do so. Hence, it is suggested to draw a good discussion on the thermodynamic data considering the results and trends they obtained and compare their data with other reported studies (if any).

2.    At present, it seems that the authors wrote this manuscript for themselves and do not want to share their data with the readers. The authors should provide supplementary file containing the equilibrium constants and other thermodynamics data they considered w.r.t temperature for each reaction in a tabular form.

3.    The major objection is that the authors mentioned that they considered 8 species only out of many other, but there is no explanation on what basis they selected or rejected the species in their study.

4.    The language and sentence construction should be improved to increase the readability of it.

Comments on the Quality of English Language

-

Author Response

(The authors gave the same response as above.)

Round 2

Reviewer 2 Report

Comments and Suggestions for Authors

The revised manuscript is suitable for its acceptance.